# Continual Transformers: Redundancy-Free Attention for Online Inference

**Lukas Hedegaard, Arian Bakhtiarnia & Alexandros Iosifidis**
Department of Electrical and Computer Engineering
Aarhus University
Aarhus, Denmark
{lhm,arianbakh,ai}@ece.au.dk

## Abstract

Transformers in their common form are inherently limited to operate on whole token sequences rather than on one token at a time. Consequently, their use during online inference on time-series data entails considerable redundancy due to the overlap in successive token sequences. In this work, we propose novel formulations of the Scaled Dot-Product Attention, which enable Transformers to perform efficient online token-by-token inference on a continual input stream. Importantly, our modifications are purely to the order of computations, while the outputs and learned weights are identical to those of the original Transformer Encoder. We validate our *Continual* Transformer Encoder with experiments on the THUMOS14, TVSeries and GTZAN datasets with remarkable results: Our *Continual* one- and two-block architectures reduce the floating point operations per prediction by up to $63\times$ and $2.6\times$, respectively, while retaining predictive performance.

## 1 Introduction

Many real-life usage scenarios such as the perception in self-driving cars and live monitoring of critical resources process a continual stream of inputs and require near-instantaneous predictions per time-step. This stands in contrast to what many common benchmarks for deep learning evaluate, namely the operation on distinct batches of data with no inter-batch relationships. Consequently, a plethora of methods have been developed (Ji et al., 2013; Carreira & Zisserman, 2017; Varol et al., 2018; Yan et al., 2018; Heidari & Iosifidis, 2021; Vaswani et al., 2017; Arnab et al., 2021; Bakhtiarnia et al., 2021b), which focus on batch-wise processing, but fail to optimise for online operation, where new information (e.g., a video frame / token) arrives at each step from a continual input stream, and future information is not available at the current time-step. We need a class of networks, which operate efficiently on *both batches of data and on continual streams*.

Accordingly, we propose a reformulation of the Transformer Encoder as a Continual Inference Network (CIN, Section 2.1) which accelerates the stream processing on time-series data, while retaining weight-compatibility. Specifically, we derive two variants of Continual Scaled Dot-Product Attention (SDA) for the cases where prior output tokes *should* and *should not* be updated after observing a new input token. Notably, our attention formulations reduce the per-step cost of SDA (Vaswani et al., 2017) from time complexity $\mathcal{O}(n^2 d)$ to $\mathcal{O}(nd)$ and memory complexity $\mathcal{O}(n^2)$ to $\mathcal{O}(nd)$ and are readily embedded into Continual Multi-Head Attention (MHA) and Continual Transformer Encoder blocks. Finally, we propose the use of Recycling Positional Encoding to accommodate progressive caching of partial attention results for continual data streams.

Due to the interdependence of SDA outputs, Continual Transformers are most efficient for shallow architectures. Shallow Transformers have many applications such as augmentations of CNNs (Touvron et al., 2021), light-weight Natural Language Processing (Cornia et al., 2020), fusion operations in multi-modal (e.g. audio-visual) settings (Chumachenko et al., 2022) and early exit branches in multi-exit architectures (Bakhtiarnia et al., 2021a;b). In our experiments[1], we validate their exceptional efficiency improvements on common benchmarks in Online Action Detection (Idrees et al., 2017) and Online Audio Classification (Tzanetakis et al., 2001).

---

[1]Source code: https://github.com/lukashedegaard/continual-transformers.

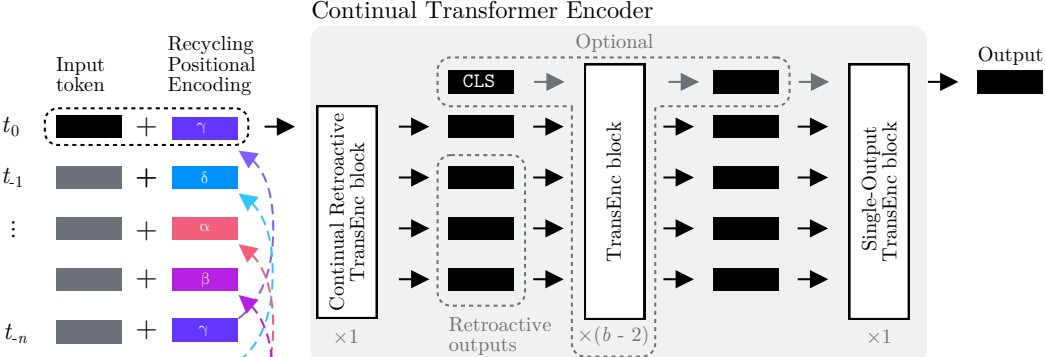

Figure 1: **Multi-block Continual Transformer Encoder with Recycling Positional Encoding.** For $b > 2$ blocks, regular Transformer Encoder blocks can be added between an initial Continual Retroactive block and a final Single-Output block. A class-token may be used after the initial block.

## 2 RELATED WORK

### 2.1 CONTINUAL INFERENCE NETWORKS

**Definition** (**Continual Inference Network**). *A Deep Neural Network, which*
- *is capable of continual step inference without computational redundancy,*
- *is capable of batch inference corresponding to a non-continual Neural Network,*
- *produces identical outputs for batch- and step inference given identical receptive fields,*
- *uses one set of trainable parameters for both batch and step inference.*

These requirements ensure that a Neural Network has broad applicability for both (offline) batch-wise inference (i.e., most research benchmarks) and online stream processing. While non-CINs can operate on streams of data by caching prior steps in a first-in first-out (FIFO) queue and aggregating them to a full (spatio-)temporal input, which is processed similarly to an offline batch, this entails computational redundancy in proportion with the sequence length. CINs perform step-wise inference without such caching and repeat computation. Uni-directional Recurrent Neural Networks are an example of Continual Inference Networks. Their default mode of operation is by time-step and they are easily applied to spatio-temporal batches of data by concatenation of the step-wise outputs. Recently, a modification to the spatio-temporal 3D convolution was proposed (Hedegaard & Iosifidis, 2021), which enables existing 3D CNNs to operate efficiently during continual inference. A similar principle was used to enhance Spatio-temporal Graph Convolutions as well (Hedegaard et al., 2022). In Section 3, we derive a CIN formulation for Transformer Encoders.

### 2.2 TRANSFORMER ARCHITECTURES

Initially proposed for sequence-to-sequence modelling in Natural Language Processing, the Transformer (Vaswani et al., 2017) has become a canonical building block in many applications of Deep Learning, including Computer Vision (Dosovitskiy et al., 2021; Arnab et al., 2021; Wang et al., 2021; Carion et al., 2020) and Audio Classification (Gong et al., 2021). Their success can be partly attributed to reduced inductive bias compared with CNNs and RNNs, which allows better adaptations when sufficiently large datasets are available; the Scaled Dot-Product Attention (SDA) maps a set of input tokens to a set of outputs without inherent preconceptions. However, this many-to-many attention exhibits quadratic growth in time and space complexity with the token count $n$ in the set.

A great deal of research has sought to improve the efficiency of Transformers (Tay et al., 2020). Block-wise or Chunking methods such as Image Transformer (Parmar et al., 2018) and Vision Transformer (Dosovitskiy et al., 2021) group up entities of a local receptive field into a single block, reducing the $\mathcal{O}(n^2)$ complexity to $\mathcal{O}(n_b^2)$, where $n_b < n$ is the number of blocks. Techniques such as sliding windows, dilation and pooling can be used to achieve a similar effect (Beltagy et al., 2020). The Reformer (Kitaev et al., 2020) reduces the complexity to $\mathcal{O}(n \log n)$ by learning group-

ings in a data-driven manner via Locality-Sensitive Hashing (LSH). A different paradigm aims to derive approximations of the self-attention matrix. Methods such as Linformer (Wang et al., 2020), Nyströmformer (Xiong et al., 2021) and Performer (Choromanski et al., 2021) reduce the complexity from $\mathcal{O}(n^2)$ to $\mathcal{O}(n)$. Unlike these efforts, our approach produces the *exact* same computational outputs for temporal sequences as the original Multi-Head Attention.

## 3 CONTINUAL TRANSFORMERS

In this work, we examine the use of Transformer Encoders for stream-processing, where we receive one token per time-step. Specifically, the query, key and value inputs constitute a continual stream of $d$-dimensional tokens and we wish to compute the outputs for each step immediately considering $n - 1$ prior tokens. We begin our exposition in Section 3.1 by considering the Scaled Dot-Product Attention (SDA) for this task. To alleviate the inefficiencies of SDA, we propose two alternative computational sequences in Section 3.2 and Section 3.3 and compare them to SDA in Section 3.4. Finally, sections 3.5-3.8 build up the full architecture, and discuss architectural considerations.

### 3.1 REGULAR SCALED DOT-PRODUCT ATTENTION

Denoting query, key, and value sequence matrices by $\boldsymbol{Q}, \boldsymbol{K}, \boldsymbol{V} \in \mathbb{R}^{n \times d}$, the regular Scaled Dot-Product Attention first defined by Vaswani et al. (2017) can be written as:

$$\text{Att}(\boldsymbol{Q}, \boldsymbol{K}, \boldsymbol{V}) = \boldsymbol{D}^{-1} \boldsymbol{A} \boldsymbol{V} \qquad \boldsymbol{A} = \exp\left(\boldsymbol{Q}\boldsymbol{K}^\top / \sqrt{d}\right) \qquad \boldsymbol{D} = \text{diag}\left(\boldsymbol{A}\mathbb{1}_n^\top\right), \qquad (1)$$

where $\boldsymbol{A}, \boldsymbol{D} \in \mathbb{R}^{n \times n}$ and $\mathbb{1}_n$ is a row-vector of $n$ ones. In each time-step, we can update $\boldsymbol{Q}$, $\boldsymbol{K}$, and $\boldsymbol{V}$ by discarding their oldest token and prepending a new one in a FIFO manner. This is a common implementation for step-wise inference, e.g. found in the FAIRSEQ library (Ott et al., 2019).

Each time-step results in $2n^2d + 2nd$ multiplications, $2n^2d - nd - n$ additions, and $n^2$ exponentiations as accounted for in Appendix A.1, which amounts to a time complexity of $\mathcal{O}(n^2d)$ and a $\mathcal{O}(n^2)$ memory complexity originating from the transient feature-map $\boldsymbol{A}$. Furthermore, a constant-sized cache of size $3(n-1)d$ is needed to store the $n - 1$ latest tokens in $\boldsymbol{Q}$, $\boldsymbol{K}$ and $\boldsymbol{V}$. We could avoid considerable redundancy by caching $\boldsymbol{Q}\boldsymbol{K}^\top$ directly. However, this comes with a memory penalty of $(n-1)^2$. Fortunately, another computational scheme can be devised.

### 3.2 CONTINUAL RETROACTIVE SCALED DOT-PRODUCT ATTENTION

We can compute $\boldsymbol{D}^{-1}\boldsymbol{A}\boldsymbol{V}$ in a step-wise manner using the latest query, key, and value steps, $\boldsymbol{q}_{\text{new}}, \boldsymbol{k}_{\text{new}}, \boldsymbol{v}_{\text{new}} \in \mathbb{R}^{1 \times d}$, alongside appropriately cached partial results. The softmax normalisation with $\boldsymbol{D}^{-1}$ can be efficiently implemented via column-aligned element-wise multiplications (denoted by $\odot$ hereafter) of a column-vector $\boldsymbol{d} = \boldsymbol{A}\mathbb{1}_n^\top$. If we cache the $n - 1$ values for the prior step tokens, i.e. $\boldsymbol{d}_{\text{mem}} = \boldsymbol{A}_{\text{prev}}^{(-n+1:-1)}\mathbb{1}_{n-1}^\top$, alongside $\boldsymbol{Q}$ and $\boldsymbol{K}$, we can define the step update as:

$$\boldsymbol{d}^{(-n+1:-1)} = \boldsymbol{d}_{\text{mem}}^{(-n+2:0)} - \exp\left(\boldsymbol{Q}_{\text{mem}}\boldsymbol{k}_{\text{old}}^\top\right) + \exp\left(\boldsymbol{Q}_{\text{mem}}\boldsymbol{k}_{\text{new}}^\top\right) \qquad (2)$$

$$\boldsymbol{d}^{(0)} = \exp\left(\frac{\boldsymbol{q}_{\text{new}}}{\sqrt{d}}\left(\boldsymbol{K}_{\text{mem}} \| \boldsymbol{k}_{\text{new}}\right)^\top\right)\mathbb{1}_n^\top, \qquad (3)$$

where $\boldsymbol{Q}_{\text{mem}}$ ($\boldsymbol{K}_{\text{mem}}$) are the $n - 1$ prior query (key) tokens, $\boldsymbol{k}_{\text{old}}$ is the key from $n$ steps ago, and $\|$ denotes concatenation of matrices along the first dimension. Negative indices indicate prior time-steps. An update for $\boldsymbol{A}\boldsymbol{V}$ can likewise be defined as a function of the $n - 1$ prior values $\boldsymbol{A}\boldsymbol{V}_{\text{mem}}$:

$$\boldsymbol{A}\boldsymbol{V}^{(-n+1:-1)} = \boldsymbol{A}\boldsymbol{V}_{\text{mem}}^{(-n+2:0)} - \exp\left(\boldsymbol{Q}_{\text{mem}}\boldsymbol{k}_{\text{old}}^\top\right)\boldsymbol{v}_{\text{old}} + \exp\left(\boldsymbol{Q}_{\text{mem}}\boldsymbol{k}_{\text{new}}^\top\right)\boldsymbol{v}_{\text{new}} \qquad (4)$$

$$\boldsymbol{A}\boldsymbol{V}^{(0)} = \exp\left(\frac{\boldsymbol{q}_{\text{new}}}{\sqrt{d}}\left(\boldsymbol{K}_{\text{mem}} \| \boldsymbol{k}_{\text{new}}\right)^\top\right)\left(\boldsymbol{V}_{\text{mem}} \| \boldsymbol{v}_{\text{new}}\right). \qquad (5)$$

Finally, we compute the Continual Retroactive Attention output in the usual manner:

$$CoRe\text{Att}(\boldsymbol{q}_{\text{new}}, \boldsymbol{k}_{\text{new}}, \boldsymbol{v}_{\text{new}}) = \boldsymbol{d}^{-1} \odot \boldsymbol{A}\boldsymbol{V}. \qquad (6)$$

An visual depiction of these update steps is provided in Appendix A.2. A time-step can now be computed with $7nd + 2n - 3d$ multiplications, $6nd + 3n - 6d - 3$ additions, and $3n - 2$ exponentials. This time complexity of $\mathcal{O}(nd)$ per step and a $\mathcal{O}(nd)$ memory complexity is a significant improvement over the prior $\mathcal{O}(n^2d)$ and $\mathcal{O}(n^2)$ complexities in Section 3.1.

### 3.3 Continual Single-Output Scaled Dot-Product Attention

Both the Regular and Continual Retroactive Dot-Product Attentions produce attention outputs for the current step, as well as $n - 1$ retroactively updated steps. In cases where retroactive updates are not needed, we can simplify the computation greatly via a Continual Single-Output Dot-Product Attention ($CoSi$Att). In essence, the regular SDA is reused, but prior values of $\mathbf{k}$ and $\mathbf{v}$ are cached between steps (as in (Ott et al., 2019)), and only the attention corresponding to a single query token $\boldsymbol{q}$ is computed:

$$CoSi\text{Att}(\boldsymbol{q}, \boldsymbol{k}_{\text{new}}, \boldsymbol{v}_{\text{new}}) = \mathbf{a}\left(\boldsymbol{V}_{\text{mem}} \parallel \boldsymbol{v}_{\text{new}}\right)/\mathbf{a}\mathbb{1}_n^\top, \qquad \mathbf{a} = \exp\left(\frac{\boldsymbol{q}}{\sqrt{d}}\left(\boldsymbol{K}_{\text{mem}} \parallel \boldsymbol{k}_{\text{new}}\right)^\top\right). \quad (7)$$

A step output is computed with $2nd + 2d$ multiplications, $2nd - d - 1$ additions, and $n$ exponentials. The time- and memory complexities remain $\mathcal{O}(nd)$ per step. Using the (leading) query $\boldsymbol{q}_{\text{new}}$ as input, the attention is purely causal. Alternatively, prior (lagging) query vectors could be cached and used as query input, though this would introduce a network delay.

### 3.4 Comparison of Scaled Dot-Product Attentions

Assuming $n - 1$ prior $\boldsymbol{q}$, $\boldsymbol{k}$ and $\boldsymbol{v}$ steps have been calculated by the Continual SDA modules, and that $\boldsymbol{Q} = (\boldsymbol{Q}_{\text{mem}} \parallel \boldsymbol{q}_{\text{new}})$, $\boldsymbol{K} = (\boldsymbol{K}_{\text{mem}} \parallel \boldsymbol{k}_{\text{new}})$, and $\boldsymbol{V} = (\boldsymbol{V}_{\text{mem}} \parallel \boldsymbol{v}_{\text{new}})$, we have the correspondence:

$$\text{Att}(\boldsymbol{Q}, \boldsymbol{K}, \boldsymbol{V})^{(t)} = CoRe\text{Att}(\boldsymbol{q}_{\text{new}}, \boldsymbol{k}_{\text{new}}, \boldsymbol{v}_{\text{new}})^{(t)} = CoSi\text{Att}(\mathbf{q}_t, \boldsymbol{k}_{\text{new}}, \boldsymbol{v}_{\text{new}}) \qquad (8)$$

Here, $\boldsymbol{q}_t$ is the $t^{\text{th}}$ row of $\boldsymbol{Q}$, i.e. $\boldsymbol{Q}^{(t)}$. During stream processing, the complexity of the Continual Retroactive SDA scales significantly more favourably that the regular SDA. For example, the floating point operations (FLOPs) are reduced by $31\times$ when $n = d = 100$ and $308\times$ when $n = d = 1000$. If retroactive output updates are not needed, the Continual Single-Output SDA reduces FLOPs by respectively $100\times$ and $1000\times$. The scaling properties are detailed in Appendix A.1.

### 3.5 Continual Multi-Head Attention

Continual Scaled Dot-Product Attentions can replace regular SDA's directly in a Multi-Head Attention (MHA). Given a new query, key, and value, $\boldsymbol{q}, \boldsymbol{k}, \boldsymbol{v}$, the Continual MHA is defined as

$$Co\text{MHA}(\boldsymbol{q}, \boldsymbol{k}, \boldsymbol{v}) = \left(\overset{h-1}{\underset{i=0}{\parallel}} Co\text{Att}(\boldsymbol{q}\boldsymbol{W}_Q^i, \boldsymbol{k}\boldsymbol{W}_K^i, \boldsymbol{v}\boldsymbol{W}_V^i)\right)\boldsymbol{W}_O, \qquad (9)$$

where $\parallel$ denotes concatenation of $h$ heads and $\boldsymbol{W}_Q^i, \boldsymbol{W}_K^i \in \mathbb{R}^{d \times d_K/h}$, $\boldsymbol{W}_V^i \in \mathbb{R}^{d \times d_V/h}$, and $\boldsymbol{W}_O \in \mathbb{R}^{d_V \times d_O}$ are projection matrices of head $i$. $Co$Att can be either $CoRe$Att or $CoSi$Att.

### 3.6 Continual Transformer Encoder

A Continual MHA block can be integrated in a Continual Transformer Encoder block as follows:

$$\boldsymbol{z} = \text{LayerNorm}\left(\boldsymbol{y} + \text{FF}(\boldsymbol{y})\right), \qquad \boldsymbol{y} = \text{LayerNorm}\left(\text{Sel}(\boldsymbol{x}) + Co\text{MHA}(\boldsymbol{x}, \boldsymbol{x}, \boldsymbol{x})\right), \qquad (10)$$

where $\boldsymbol{x}$ corresponds to the newest step input and $\text{Sel}(\cdot)$ selects a single (last) token of $\boldsymbol{x}$ if *CoSi*MHA is used, or selects all tokens otherwise. $\text{FF}(\cdot)$ is a two-layer feed-forward network with weights $\boldsymbol{W}_1, \boldsymbol{W}_2$, biases $w_1, w_2$, and a activation function $\sigma(\cdot)$, i.e. $\text{FF}(\boldsymbol{x}) = \sigma(\boldsymbol{x}\boldsymbol{W}_1 + w_1)\boldsymbol{W}_2 + w_2$. Aside from the residual selection, this is identical to common Transformer Encoder implementations (Vaswani et al., 2017; Dosovitskiy et al., 2021).

### 3.7 Recycling Positional Encoding

Since a Transformer Encoder does not provide positional bias, it is common to augment a token $\boldsymbol{x}_i$ with a positional encoding $\boldsymbol{p}$, i.e. $\tilde{\boldsymbol{x}}_i = \boldsymbol{x}_i \circ \boldsymbol{p}_i$, where $\circ$ could be addition or concatenation. In regular Transformers, the index $i$ denotes a position in a sequence rather than a position in time. However, this static positional assignment is problematic in the context of continual inference; the last token at time $t = 0$ will be the next-to-last token at time $t = 1$, and thus in need of a different

positional encoding than in the prior time-step. Instead, CINs require dynamic positions. There have been multiple prior works (Shaw et al., 2018; Huang et al., 2019; Dai et al., 2019) which create relative encodings by augmenting the SDA with positional offsets $\boldsymbol{P}$ between query and keys, i.e. $\boldsymbol{A} = \exp(\boldsymbol{Q}\boldsymbol{K}^\top/\sqrt{d} + \boldsymbol{P})$. While a similar modification to the continual attentions is possible, it is incompatibile with the regular SDA in Eq. (1). Instead, we use a *Recycling Positional Encoding* (RPE), which lets the positional encoding follow each token in time and recycles old encodings:

$$\tilde{\boldsymbol{x}}_t = \boldsymbol{x}_t + \boldsymbol{p}_{\tau_t}, \qquad \tau_t = (\tau_{t-1} + 1) \bmod T, \tag{11}$$

where $T$ is the number of encodings. While RPE does not specify relative encodings explicitly, the absolute positional interpretation of each token changes dynamically when a new token arrives. In practice, the network learns relative, shift-invariant positional information by training with random $\tau_0$ for each batch. Random shifts during training were recently explored in (Kiyono et al., 2021; Likhomanenko et al., 2021; Dehghani et al., 2019) as well. RPE can use either learned or predefined encodings. In the latter case, Cyclic Positional Encoding (Ma et al., 2021), a sinusoidal encoding inspired by Gray code, is a good fit. If we reuse the encoding immediately after an old token has "slided out", i.e. $T = n$, a token will have the same positional encoding relative to another whether it was $m$ steps older or $n - m$ steps newer. The positional ambiguity can be avoided by extending the number of positional tokens to $T = 2n - 1$. We explore both options in Section 4.1.2.

### 3.8 ARCHITECTURAL CONSIDERATIONS

**Block count** In Section 3.4, we observed an exact correspondence between the results of the continual and regular SDA layers. However, the correspondence does not necessarily hold for stacked layers. Consider the result of stacking two Continual Single-Output Transformer Encoder blocks. While the first block outputs a step $t$ that is identical to that in a corresponding regular block, the second block would have been initialised with prior step-wise inputs, which were the result of prior input windows instead of the current one; the correspondence would not hold. Though it is not convertible to/from a regular Transformer Encoder, the stacked Single-Output Transformer Encoder architecture has the merit of efficiency. This was exploited in Transformer-XL (Dai et al., 2019). Given a single step input, the Continual Retroactive Transformer Encoder block produces output tokens corresponding to the entire observed sequence inside the window. Due to this one-to-many input-output mapping, it is not possible to stack multiple such layers. Nevertheless, it can be used in conjunction with a Continual Single-Output Transformer Encoder with optional regular Transformer Encoder blocks in between as illustrated in Fig. 1. The Regular Transformer Encoder blocks in the resulting architecture have a significantly larger computational complexity than the Continual Retroactive and Single-Output blocks. Consequently, we recommend that Continual Transformer Encoders be used primarily in lightweight architectures with one or two blocks unless compatibility with non-continual Transformers is not required and only Single-output blocks are used.

**Class token** It is common to add a class token as input to transformers (Devlin et al., 2019; Dosovitskiy et al., 2021), which accumulates information from other tokens prior to classification. However, it cannot be used naïvely with CINs, as this would effectively double the number of input steps. In practice, it can be employed in Continual multi-block Transformer Encoders as input to the second block (see Fig. 1), but this placement limits class token interaction with downstream layers. It can also be used for one-block Transformer Encoders if the value token is omitted as input.

**Peak memory reduction trick** The FLOPs for $\mathrm{Att}(\boldsymbol{Q}, \boldsymbol{K}, \boldsymbol{V})$ are exactly $n$ times those of $Co Si \mathrm{Att}(\mathbf{q}, \boldsymbol{k}_{\mathrm{new}}, \boldsymbol{v}_{\mathrm{new}})$. Comparing their memory complexity, the regular SDA is $\mathcal{O}(n^2)$, while the Single-output SDA is $\mathcal{O}(nd)$. In practical applications where system memory is limited, we may thus reduce the maximum memory requirement of the computational device at inference by up to $d/n$ (assuming $n \gg d$) by computing each row of the attention individually. However, this may reduce throughput due to reduced parallelism.

## 4 EXPERIMENTS

We provide case studies within two perception disciplines, namely Online Action Detection (Section 4.1) and Audio Classification (Section 4.2). In each case, we will start with a brief overview of the field, followed by experiments and results.

## 4.1 ONLINE ACTION DETECTION

Online Action Detection (OAD) (De Geest et al., 2016) entails the per-frame classification of human actions in a video stream as they happen without the ability to change prior predictions nor to use future information. This is fundamentally more restrictive than Temporal Action Localisation, where the whole video clip is processed before start and end frames of an action are determined (Shou et al., 2016; Xu et al., 2017; Shou et al., 2017; Wu et al., 2019).

The dominant design in OAD works at the time of writing is to employ a two-stream Convolutional Neural Network as backbone for frame-wise feature extraction with RGB images as inputs in one stream and Optical Flow fields in the other (Gao et al., 2017; Xu et al., 2019; Eun et al., 2020; Wang et al., 2021; Xu et al., 2021)[2]. On top of these, OAD methods encode temporal information and perform predictions per time-step, e.g. by means of RNNs (Gao et al., 2017; Xu et al., 2019; Eun et al., 2020) or Transformers (Wang et al., 2021; Xu et al., 2021). Alongside the action detection for the current frame, an action anticipation task may be learned in parallel by means of decoder structures, as this has been found to improve the primary OAD task.

Unlike RNNs, an output update for the regular SDA in a Transformer block cannot be naïvely computed for a single step by feeding successive video frames. Instead, prior step features must be cached, re-loaded and re-processed by the Transformer in each step in correspondence with a predefined window-size of prior steps. As laid out in Section 3.8, Continual Transformers are especially efficient when either one or two Continual Transformer Encoder blocks are used. Accordingly, we start our experiments with a set ablation studies to simplify a recent transformer-based architecture, the OadTR (Wang et al., 2021). We further investigate the impact of ablating class token position and the use of Recycling Positional Encoding and compare different RPE schemes for Continual Transformers. Finally, we evaluate our configurations on two widely used OAD datasets, THUMOS14 (Idrees et al., 2017) and TVSeries (De Geest et al., 2016).

### 4.1.1 EXPERIMENTAL SETUP

The THUMOS14 dataset (Idrees et al., 2017) for OAD has 200 and 213 validation and test videos, respectively, with frame-level class annotations across 20 classes. As in prior OAD works, the model is trained on the validation set and evaluated on the test set. Similar to Wang et al. (2021) we use pre-extracted features from a two-stream Temporal Segment Network (TSN) (Wang et al., 2019) trained on ActivityNet v1.3 (Heilbron et al., 2015) or Kinetics-400 (Carreira & Zisserman, 2017).

For TVSeries (De Geest et al., 2016), the network learns on the train and validations sets (20 videos) and evaluates on the test set (7 videos) as in (Wang et al., 2021). RGB and Optical Flow features were extraced using an MMAction2 (Contributors, 2020) pipeline with ActivityNet v1.3 (Heilbron et al., 2015) and Kinetics-400 (Carreira & Zisserman, 2017) pretrained TSN ResNet-50 (He et al., 2016) backbones. This is similar to the feature extraction process used by LSTR (Xu et al., 2021).

Following Wang et al. (2021), we use a batch size of 128, sequence length 64, initial learning rate $10^{-4}$ with a factor ten reduction each epoch, alongside weight decay $10^{-4}$, and dropout with probability 0.1. We report results using two epochs of training on a Nvidia RTX2080 Ti GPU. We track mean Average Precision (mAP) for THUMOS14 and calibrated mean Average Precision (cmAP) (De Geest et al., 2016) for TVSeries, alongside FLOPs per prediction and parameters of the OAD module (feature extraction excluded). We report the mean $\pm$ standard deviation over five runs.

### 4.1.2 ABLATION STUDIES

**Removing the Decoder** As a first step to make an efficient Continual OadTR, we remove the decoder blocks used for action anticipation, which has a large impact on computational efficiency and the ease of transformation to a Continual Inference Network. The first two lines of Table 1a present the results of the removal. Contrary to the observations of Wang et al. (2021), we did not find any drop in accuracy when excluding the decoder. We do, however, gain a large reduction in FLOPs and model size; they were reduced to 58% and 30%, respectively. Given these computational improvements, we exclude the decoder in subsequent experiments.

---

[2]The feature extraction commonly used in Online Action Detection (OAD) works is in itself quite computationally costly. We consider the optimisation of the backbone as orthogonal future work and will follow the same feature extraction procedure as other OAD works at this time.

Table 1: **Ablation experiments** on THUMOS14 with TSN-Anet features. **Best** metrics are highlighted. '-' indicates that a particular feature was not used.

(a) **Class token** variations with OadTR. `CLS` pos. is the encoder block into which `CLS` is input.

| Enc. blocks | Dec. | CLS pos. | mAP (%) | FLOPs (M) | Params (M) |
|---|---|---|---|---|---|
| 3 | ✓ | 1 | $57.0_{\pm0.5}$ | 2445.6 | 74.7 |
| 3 | - | 1 | $57.0_{\pm0.4}$ | 1430.6 | 22.2 |
| 3 | - | 2 | $\mathbf{57.3_{\pm0.7}}$ | 1423.5 | 22.2 |
| 3 | - | 3 | $56.7_{\pm0.6}$ | 1417.2 | 22.2 |
| 3 | - | - | $56.8_{\pm0.3}$ | **1410.9** | 22.2 |
| 2 | - | 1 | $56.5_{\pm0.5}$ | 1020.7 | 15.9 |
| 2 | - | 2 | $\mathbf{56.7_{\pm0.3}}$ | 1014.5 | 15.9 |
| 2 | - | - | $56.6_{\pm0.3}$ | **1008.1** | 15.9 |
| 1 | - | 1 | $\mathbf{57.1_{\pm0.6}}$ | 611.7 | 9.6 |
| 1 | - | - | $56.3_{\pm0.2}$ | **605.5** | 9.6 |

(b) **Positional encodings** variations for *Co*OadTr.

| Enc. blocks | Re-cycling | Learn | Pos. tokens | mAP (%) | FLOPs (M) | Params (K) |
|---|---|---|---|---|---|---|
| 2 | - | ✓ | $n$ | $45.3_{\pm0.9}$ | 410.9 | 15832 |
| 2 | ✓ | ✓ | $n$ | $56.4_{\pm0.3}$ | 410.9 | 15832 |
| 2 | ✓ | ✓ | $2n-1$ | $56.0_{\pm0.5}$ | 410.9 | 15897 |
| 2 | ✓ | - | $n$ | $55.8_{\pm1.0}$ | 410.9 | **15767** |
| 2 | ✓ | - | $2n-1$ | $\mathbf{56.8_{\pm0.4}}$ | 410.9 | **15767** |
| 1 | - | ✓ | $n$ | $44.0_{\pm0.8}$ | 9.6 | 9535 |
| 1 | ✓ | ✓ | $n$ | $55.6_{\pm0.3}$ | 9.6 | 9535 |
| 1 | ✓ | ✓ | $2n-1$ | $55.6_{\pm0.3}$ | 9.6 | 9599 |
| 1 | ✓ | - | $n$ | $54.4_{\pm1.8}$ | 9.6 | **9469** |
| 1 | ✓ | - | $2n-1$ | $\mathbf{56.1_{\pm0.7}}$ | 9.6 | **9469** |

**(Re)moving the Class token** Class tokens should not be input naively to the first Transformer Encoder layer of a CIN (see Section 3.8). Accordingly, we ablate its use and position. In cases where it is removed, we predict on the token corresponding to the last input token. The results of varying `CLS` pos are noted in Table 1a. For the one-block architecture, the removal came with noticeable drop in mAP, while the two-block architecture saw small improvements when removing or introducing the class token later. For the three block model, the use of class tokens in block two achieved the highest mAP. Though it is commonly accepted, that class tokens should be introduced alongside other inputs in the first block, our results indicate that they can accumulate sufficient information with only one or two blocks, and that later stage introduction may work better in some applications. In general, the achieved mAP when varying `CLS` pos. and number of blocks are very similar to one another, while (re)moving the class token and reducing the block size both reduce computational complexity. This encourages the use of shallow Transformer Encoders over deeper ones as well as the removal of class tokens, as we do in the following experiments.

**Positional Encodings** We can transfer parameters from the simplified one- and two block OadTR to the corresponding Continual architecture, *Co*OadTR. Here, the one block version (*Co*OadTR-b1) uses *CoSi*MHA, and the two block model (*Co*OadTR-b2) uses *CoRe*MHA in the first block and Single-output MHA in the second. However, a regular positional encoding is not suited for continual inference (see Section 3.7). We evaluate the performance of using non-continual encodings for continual inference, as well as of our proposed Recycling Positional Encodings with fixed or learned parameters. In addition, we explore the impact of extending the number of tokens from $n$ to $2n-1$ to avoid positional ambiguity. As seen in Table 1b), non-continual encoding used in the continual setting result in severe mAP drop. Recycling Positional Encodings alleviate this. Comparing learned and fixed encodings, we find the learned encodings to work better when the number of encoding tokens corresponds to the sequence length $n$ and the fixed encoding to work best when positional ambiguity is alleviated by extending the number of tokens to $2n-1$. Fixed encoding with $2n-1$ tokens works best overall and is employed in subsequent experiments unless stated otherwise. There is no difference in FLOPs for either strategy, and the difference in parameter count is negligible.

### 4.1.3 COMPARISON WITH PRIOR WORKS

We evaluate the (*Co*)OadTR architectures on THUMOS14 and TVSeries with two sets of features as described in Section 4.1.1. Since no prior OAD works have reported complexity metrics, we measured the FLOPs for TRN (Xu et al., 2019) based on the publicly available source code to serve as a point of reference. The results of this benchmark are presented in Table 2 and Fig. 2. OadTR and our simplified (continual) one-block (b1) and two-block (b2) versions without decoder and class tokens generally achieve competitive precision in comparison with prior works, surpassing all but OadTR and LSTR. On THUMOS14, our reproduced OadTR results are slightly lower than originally reported (Wang et al., 2021)[3], whereas achieved TVSeries results are higher[4]. The (*Co*)OadTR-b#

---

[3]The reported 58.3% on THUMOS14 could not be reproduced using their publicly available code.

[4]We attribute our higher mcAP to differences in the feature extraction pipeline.

Table 2: **Online Action Detection** results. FLOPs per prediction are noted for inference on THUMOS14. The **best** and *next-best* metrics are highlighted.

| Model | Feat. | THUMOS14 mAP (%) | TVSeries mcAP (%) | FLOPs (M) |
|---|---|---|---|---|
| RED (Gao et al., 2017) | A.Net | 45.3 | 79.2 | - |
| TRN (Xu et al., 2019) | | 47.2 | 83.7 | 1387.5 |
| FATS (Kim et al., 2021) | | 51.6 | 81.7 | - |
| IDN (Eun et al., 2020) | | 50.0 | 84.7 | - |
| TFN (Eun et al., 2021) | | 55.7 | 85.0 | - |
| LSTR (Xu et al., 2021) | | **65.3** | 88.1 | - |
| OadTR (Wang et al., 2021) | | *58.3* | 85.4 | 2445.6 |
| OadTR[†] | | 57.0±0.5 | **88.6±0.1** | 2445.6 |
| OadTR-b2[†] | | 56.6±0.3 | *88.3±0.2* | 1008.1 |
| OadTR-b1[†] | | 56.3±0.2 | 88.1±0.1 | 605.5 |
| *Co*OadTR-b2 (ours) | | 56.8±0.4 | 87.7±0.6 | *410.9* |
| *Co*OadTR-b1 (ours) | | 56.1±0.7 | 87.6±0.7 | **9.6** |
| TRN (Xu et al., 2019) | Kin. | 62.1 | 86.2 | 1462.0 |
| FATS (Kim et al., 2021) | | 59.0 | 84.6 | - |
| IDN (Eun et al., 2020) | | 60.3 | 86.1 | - |
| PKD (Zhao et al., 2020) | | 64.5 | 86.4 | - |
| LSTR (Xu et al., 2021) | | **69.5** | **89.1** | - |
| OadTR (Wang et al., 2021) | | *65.2* | 87.2 | 2513.5 |
| OadTR[†] | | 64.2±0.3 | **88.6±0.1** | 2513.5 |
| OadTR-b2[†] | | 64.5±0.5 | 88.3±0.2 | 1075.7 |
| OadTR-b1[†] | | 63.9±0.5 | 88.1±0.1 | 673.0 |
| *Co*OadTR-b2 (ours) | | 64.4±0.1 | 87.6±0.7 | *411.9* |
| *Co*OadTR-b1 (ours) | | 64.2±0.4 | 87.7±0.4 | **10.6** |

[†]Using official source code or modifications there-off.

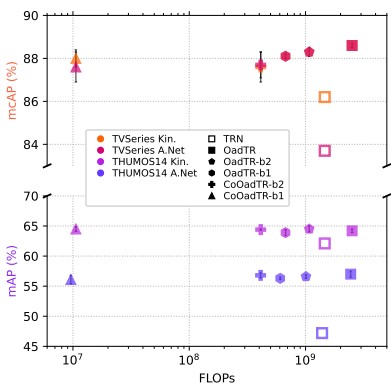

Figure 2: **Visual comparison** of OAD methods on THUMOS14 and TVSeries for backbones trained on ActivityNet 1.3 and Kinetics-400.

architecture largely retain precision and allow significantly reduced FLOPs per prediction. Our proposed continual variants *Co*OadTR-b1 and *Co*OadTR-b2 reduce FLOPs by **255×** and **6.1×**, respectively, compared to OadTR, while either achieving the same performance or conceding no more than one percentage point. On average, continual and non-continual (*Co*)OadTR-b# models achieve similar mAP on THUMOS14, while OadTR-b# have slightly higher mcAP on TVSeries. We attribute these discrepancies to differences in positional encoding. All in all, the *Co*OadTR-b# models provide far-superior computational efficiency to prior works, achieving state-of-the-art performance/efficiency trade-offs by a large margin.

### 4.1.4 AUDIO-VISUAL ONLINE ACTION DETECTION

To showcase the validity of our method in audio-visual settings as well, we explore the addition of audio-features to the Online Action Detection task on THUMOS14. As described in Section 4.2, audio-features are extracted using Mel spectrograms and an AudioSet pre-trained VGGish network (Hershey et al., 2017) (output of the penultimate layer) on 1.0 second windows with a step size of 0.2 seconds to match the 5.0 FPS sampling rate of the video features.

The audio-features by themselves do not provide enough signal to reach good Online Action Detection performance (yielding only 6.7% mAP with an OadTR network). When concatenated with RGB and Flow they do provide a modest improvement as seen in Table 3. On average, this amounts to +0.6% mAP when combined with ActivityNet features and +0.5% mAP when used with Kinetics-400 features with shallower models enjoying the largest improvements.

### 4.2 AUDIO CLASSIFICATION

### 4.2.1 BACKGROUND

Audio Classification is the categorisation of audio waveforms. Though waveform sequences can be used directly (Lee et al., 2017), it is common to first convert them to spectrograms. Mel spectrograms are obtained by a nonlinear transformation of a frequency scale (Stevens et al., 1937), which is designed based on empirical knowledge about the human auditory system (Choi et al., 2016). By employing spectrograms, audio classification can be approached in the same way as image classification (Palanisamy et al., 2020).

Table 3: **Audio-Visual** result, THUMOS14.

| Model | Feat. | mAP (%) | FLOPs (M) |
|---|---|---|---|
| OadTR | | $57.6_{\pm0.6}$ | 2714.9 |
| OadTR-b2 | A.Net | $57.5_{\pm0.5}$ | 1277.0 |
| OadTR-b1 | + | $57.4_{\pm0.4}$ | 874.1 |
| *Co*OadTR-b2 | AudioSet | $56.5_{\pm1.1}$ | *415.0* |
| *Co*OadTR-b1 | | $56.8_{\pm0.5}$ | **13.8** |
| OadTR | | $64.4_{\pm0.4}$ | 2781.9 |
| OadTR-b2 | Kin. | $65.0_{\pm0.4}$ | 1344.1 |
| OadTR-b1 | + | $64.5_{\pm0.4}$ | 941.2 |
| *Co*OadTR-b2 | AudioSet | $64.7_{\pm0.8}$ | *416.0* |
| *Co*OadTR-b1 | | $64.8_{\pm0.3}$ | **14.8** |

Table 4: **Audio Classification** results for GTZAN.

| Method | Pos. Enc. | Acc. (%) | FLOPs (M) | Par. (K) |
|---|---|---|---|---|
| Maj. Voting | - | 92.0 | - | **0** |
| Trans-b2 | learned | $95.0_{\pm0.6}$ | 47.4 | 509 |
| Trans-b1 | learned | $93.8_{\pm0.8}$ | 15.2 | *286* |
| *Co*Trans-b2 | fixed | $94.4_{\pm1.0}$ | 27.0 | 509 |
| *Co*Trans-b1 | learned | $93.2_{\pm1.1}$ | *0.3* | *286* |

### 4.2.2 EXPERIMENTS

We conduct experiments on the Music Genre Classification dataset GTZAN (Tzanetakis & Cook, 2002). It consists of 100 30-second clips for each of ten music genres. Each audio clip is sampled at 22,050 Hz. Since there are no predefined splits for GTZAN, we randomly select 10% of the data for validation and 10% for testing. The input is transformed to a temporal sequence by sliding a one-second window over each 30-second clip with a slide step size of 250ms, leading to 120 one-second clips. These are subsequently converted to Mel spectrograms. We then fine-tune a VGGish network, pre-trained on AudioSet (Hershey et al., 2017) and use the penultimate layer for feature extraction. A batch size of 64 and the Adam optimizer (Kingma & Ba, 2015) are used with an initial learning rate of $10^{-4}$. The learning rate is reduced by a factor of $0.6$ on plateau with a tolerance of two epochs, and an early stopping mechanism, where a maximum of 100 epochs are allowed. The VGGIsh base-network attains an accuracy of 86.1% on the dataset of one-second clips with 72.1M parameters and 864.7M FLOPs. Subsequently, the audio features are passed to a (Continual) Transformer Encoder which has 16 attention heads, an embedding dimension of 192 and an MLP dimension of 384. The Transformer Encoder is trained on the whole temporal sequence using a batch size of 32 and the AdamW optimizer (Loshchilov & Hutter, 2019) with a learning rate of $10^{-5}$ and a weight decay of $10^{-4}$ for 50 epochs. Since the Transformer Encoder is trained on entire 30-second clips, there are less data points available for this training. Accordingly, the size of the validation set is increased to 18%. All audio classification training procedures were carried out on a single Nvidia RTX 2080 Ti GPU. Table 4 presents the accuracy and efficiency of regular and Continual Transformers during online inference. As a baseline, we also include the result of majority voting among the clips to classify the entire sequence. The Continual Transformers obtain similar accuracy as regular a Transformers while consuming $1.76\times$ less FLOPs when using two blocks and $51.5\times$ less FLOPs when using one Transformer Encoder block.

## 5 CONCLUSION

In this work, we presented Continual Transformers, a redundancy-free reformulation of Transformers tailored for online inference. Central to the Continual Transformer are the Continual Retroactive and Single-Output Attention operations, which produce outputs identical to the original Scaled Dot-Product Attention for continual input sequences, while greatly reducing the time and memory complexity per prediction. The applicability of Continual Transformer architectures was experimentally validated in Online Action Detection and Online Audio Classification settings, observing upwards of multiple orders of magnitude reduction in time complexity for lightweight architectures at modest accuracy concessions. Continual Transformers constitute an algorithmic innovation, which could make possible hitherto unseen precision, speed, and power efficiency in online inference use-cases. With applications spanning enhanced perception and reactivity of robots and autonomous vehicles, weather forecasting, price prediction and surveillance, we hope it will be used for the common good.

### ACKNOWLEDGMENTS

This work has received funding from the European Union's Horizon 2020 research and innovation programme under grant agreement No 871449 (OpenDR).

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

## A  APPENDIX

### A.1  SCALING PROPERTIES OF CONTINUAL AND REGULAR MULTI-HEAD ATTENTION

A detailed account for the floating point operations involved in computing Regular-, Continual Retroactive-, and Single-output Scaled Product Attentions is given in Tables 5, 6, and 7.

Fig. 3 illustrates the scaling of FLOPs and memory footprint with increasing sequence length $n$ and embedding dimension $d$. Here, the Continual Retroactive and Single-Output SDAs spend significantly less FLOPs than the Regular SDA, which scales $\mathcal{O}(n^2)$ as opposed to $\mathcal{O}(nd)$ the continual variants. The Continual Single-Output SDA reduces memory footprint for all value combinations, and the Continual Retroactive SDA does so when $n \gtrsim d$.

Table 5: **Floating Point Operations** for the Scaled Dot-Product Attention in Eq. (1). $\boldsymbol{D}^{-1}(\cdot)$ can be efficiently computed as element-wise multiplication with $\boldsymbol{AV}$.

|          | Mul.         | Add         | Exp   |
|----------|--------------|-------------|-------|
| Eq. (1.1) | $n^2d + nd$ | $nd(n-1)$   | $0$   |
| Eq. (1.2) | $n^2d + nd$ | $n^2(d-1)$  | $n^2$ |
| Eq. (1.3) | $0$         | $n(n-1)$    | $0$   |

Table 6: **Floating Point Operations** for the Continual Retroactive Dot-Product Attention in Eqs. (2) to (6). The outputs of the exponentials in Eq. (2) and Eq. (3) can be reused in Eq. (4) and Eq. (5) respectively, and are omitted in the count.

|          | Mul.           | Add                    | Exp        |
|----------|----------------|------------------------|------------|
| Eq. (2)  | $2(n-1)d$      | $2(n-2)d + 2(n-1)$     | $2(n-1)$   |
| Eq. (3)  | $nd + n + d$   | $nd + (n-1) + d$       | $n$        |
| Eq. (4)  | $2(n-1)d$      | $2(n-1)d$              | $0$        |
| Eq. (5)  | $nd$           | $(n-1)d$               | $0$        |
| Eq. (6)  | $nd + n$       | $0$                    | $0$        |

Table 7: **Floating Point Operations** for the Continual Single-Output SDA in Eq. (7).

|          | Mul.      | Add                  | Exp  |
|----------|-----------|----------------------|------|
| Eq. (7.1) | $nd + d$ | $(n-1)d + n - 1$     | $0$  |
| Eq. (7.2) | $nd + d$ | $n(d-1)$             | $n$  |

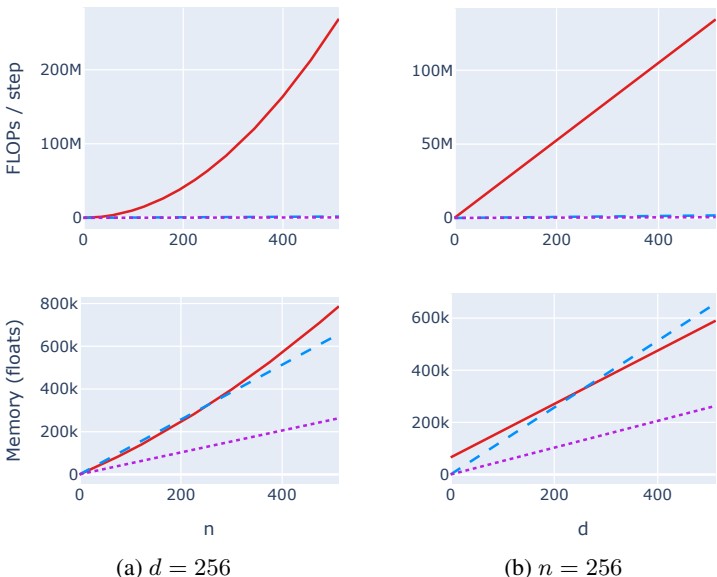

(a) $d = 256$      (b) $n = 256$

Figure 3: **FLOPs/step and memory footprint** for Regular, Continual Retroactive, and Continual Single-Output Scaled Dot-Product Attention at varying sequence length $n$ and embedding dimension $d$. Column (a) has $d$ fixed to 256; Column (b) has $n$ fixed to 256.

## A.2 SUPPLEMENTAL VISUALISATIONS

For the visually inclined, we supply a complementary graphical depictions of the Continual Retroactive SDA corresponding to Eqs. (2) to (6) in Fig. 4 and the Single-Output SDA in Eq. (7) in Fig. 5.

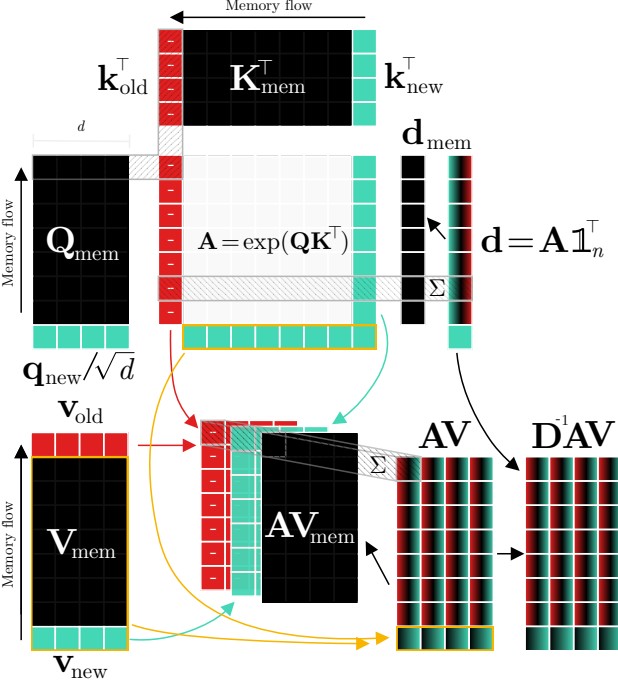

Figure 4: **Continual Retroactive Dot-Product Attention**. The query ($Q$), key ($K$), and value ($V$) matrices are aggregated over time by caching the step vectors $q_{new}$, $k_{new}$, and $v_{new}$ in each their FIFO queue (denoted by $\square_{mem}$). During each step, only the entries of $A$ associated with $q_{new}$, $k_{new}$ and the oldest $K$ step, $k_{old}$ are computed. The diagonal entries of the row-normalisation matrix $D$ as well as the $AV$ can be updated retroactively by subtracting features corresponding to $k_{old}$ and adding features related to $k_{new}$ to the cached outputs of the previous step, $D_{mem}$ and $AV_{mem}$, respectively.

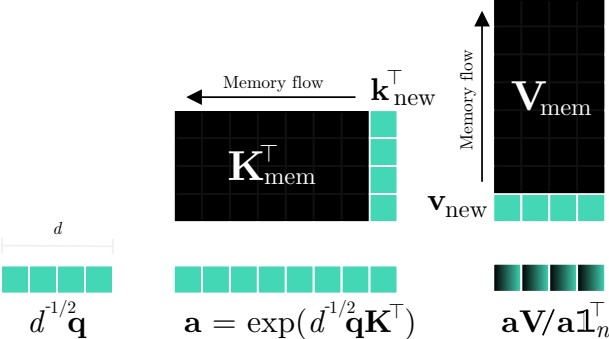

Figure 5: **Continual Single-Output Dot-Product Attention**. The key ($K$) and value ($V$) matrices are aggregated over time by caching the step vectors $k_{new}$ and $v_{new}$ in a FIFO queue. During each step, only the attention output associated with $q$ is computed.

A schematic illustration of the Audio Classification experiments architecture is depicted in Fig. 6.

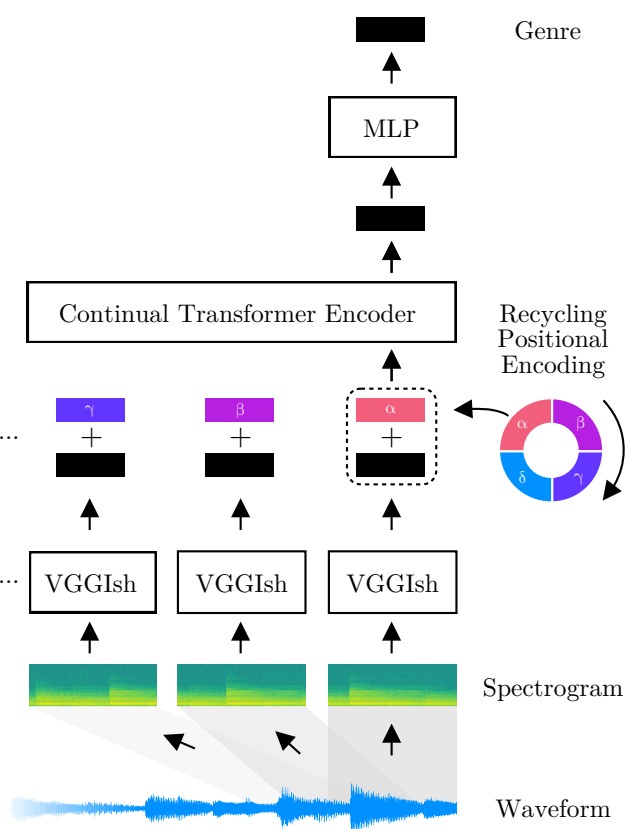

Figure 6: **Audio Classification Architecture**.

