# OpenReview forum: "Continual Transformers: Redundancy-Free Attention for Online Inference"
_ICLR.cc/2023/Conference — ICLR 2023 poster_

### Official Review · Reviewer_RfrZ · 2022-10-24

**Confidence:** 3
**Correctness:** 4
**Technical Novelty And Significance:** 3
**Empirical Novelty And Significance:** 3
**Recommendation:** 6

**Clarity, Quality, Novelty And Reproducibility:**

The paper proposes a novel approach for the transformer efficiency on continual data streams. But the writing is not so clear. I think with a revision of the text, it can be a good paper.

**Strength And Weaknesses:**

Strengths:
- The proposed approach is novel and provides an efficient and simple solution for continual data streams.
- It is a significant contribution to reduce the time and memory complexity per prediction while keeping the performance on par.
- To the best of my knowledge, the statements are correct.

Weaknesses:
- I had difficulties to follow the paper. I think a re-organization or restructuring of the sections is needed. At least in the beginning of the Section 3, there could be an outline of the section, so we can follow the story in the section.
- Besides, the experiment results could be analysed in more detail. It is difficult to understand whether the proposed method's accuracy is sufficiently close to the baselines or what are the outcomes of the experiments.

**Summary Of The Paper:**

In this paper, they propose a new formulation of scaled dot-product attention for token by token inference on continual stream. That reduces the time complexity from O(n^2 d) to O(n d) while keeping the outputs and weights identical to the original Transformer outputs and weights.

**Summary Of The Review:**

I think the paper's contribution is important for community, it only requires a more clear presentation.

---

> ### Author Response · Authors · 2022-11-09
> **Point-to-point response to comments from Reviewer RfrZ**
>
> > I had difficulties to follow the paper. I think a re-organization or restructuring of the sections is needed. At least in the beginning of the Section 3, there could be an outline of the section, so we can follow the story in the section.
>
> Thank you for your suggestions. We have followed your proposal to add an outline to Section 3 to introduce narrative better. The introductory text now reads as follows: "In this work, we examine the use of Transformer Encoders for stream-processing, where we receive one token per time-step. Specifically, the query, key and value inputs constitute a continual stream of d-dimensional tokens and we wish to compute the outputs for each step immediately considering n − 1 prior tokens. We begin our exposition in Section 3.1 by considering the Scaled Dot-Product Attention (SDA) for this task. To alleviate the inefficiencies of SDA, we propose two alternative computational sequences in Section 3.2 and Section 3.3 and compare them to SDA in Section 3.4. Finally, sections 3.5-3.8 build up the full architecture, and discuss architectural considerations.”
>
> > Besides, the experiment results could be analysed in more detail. It is difficult to understand whether the proposed method's accuracy is sufficiently close to the baselines or what are the outcomes of the experiments.
>
> As seen in Table 2, the predictive performance is comparable between the baselines (e.g. OadTR) and our proposed CoOadTR-b{1,2} models. Not only is the performance of CoOadTR-b{1,2} is within -0.5 to 0.0 mAP of OadTR (which is expected considering the greatly simplified model architectures), their efficiency is far-superior, achieving 255x and 6.1x reductions in FLOPs, respectively. We have highlighted these metrics in 4.1.3.
>
> Comparing equivalent model architectures (e.g. OadTR-b1 with CoOadTr-b1), you will see that performance is very close (-0.2 to +0.3 mAP), while relying on 63x less FLOPs.  This is stated in Section 4.1.3: "On average, continual and non-continual (Co)OadTR-b# models achieve similar mAP on THUMOS14, while OadTR-b# have slightly higher mcAP on TVSeries”.
> In reference to the other baselines (RED, TRN, FATS, IDN, TFN, LSTR), the achieved mAP/mcAP is also very competitive despite the significantly lower FLOPs needed per prediction.
>
> We have added the following to Sec 4.1.3 as a conclusion to the experiments: “All in all, the CoOadTR-b# models provide far-superior computational efficiency to prior works, achieving state-of-the-art performance/efficiency trade-offs by a large margin.“
> Unfortunately, the page limitation prohibits us from expanding our descriptions and comments on the experimental results further.

---

> > ### Comment · Reviewer_RfrZ · 2022-11-20
> > **Acknowledge**
> >
> > Thank you for your responses. After reading the other reviews and the rebuttal, I keep my score as is.

---

### Official Review · Reviewer_DN9Y · 2022-10-24

**Confidence:** 4
**Correctness:** 4
**Technical Novelty And Significance:** 3
**Empirical Novelty And Significance:** 3
**Recommendation:** 8

**Clarity, Quality, Novelty And Reproducibility:**

The paper is well written and is clear to follow. However, I found that there is a lack of motivation behind some points, such as the motivation for using Recycling Positional Encoding, and architecture modification. This makes it hard to draw conclusions from experiments.

**Strength And Weaknesses:**

Strength:
- The paper is well presented and the proposed architecture is explained in detail.
- The paper is well positioned with respect to prior work
- The experiments show that the proposed model reduces the time and memory complexity per predictions.

Weaknesses:
- The motivation in some aspects is not clear. For example, the authors suggest the use of Recycling Positional Encoding to accommodate progressive caching of partial attention results for continual data streams. In section 3.7 they said that “There have been multiple prior works (Shaw et al., 2018; Huang et al., 2019; Dai et al., 2019) which create relative encodings by augmenting the SDA with positional offsets between query and keys. While such a modification to the continual attentions is possible, it hinders compatibility with the regular SDA.`` Why is Recycling Positional Encoding more helpful here and how does the previous model hinder the compatibility?
- The authors said that the proposed model works better with shallow architecture; is it still applicable to deeper architecture?
- How is the information accumulated from other tokens prior to classification?
- Uni-directional Recurrent Neural Networks are an example of Continual Inference Networks. How is the proposed model improved upon the Uni-directional Recurrent Network? For example, in SDA prior step features must be cached, re-loaded, and re-processed by the transformer in each step in correspondence with a predefined window-size of prior steps. Don't the authors think this is very complicated compared with RNN?
- The experiments show that the proposed model performs well in time and memory reduction but it does not outperform SOTA. Can the authors explain more about that?


**Summary Of The Paper:**

In this paper, the authors propose a novel transformer model of the Scaled-Dot-Product Attention. The authors claim that the proposed continual transformers will accelerate the stream processing of transformer architectures on time-series data. The continual transformer architectures were experimentally validated in Online Action Detection and Online Audio Classification.

**Summary Of The Review:**

Overall, I think this is an interesting paper even though the reported results are slightly lower than SOTA. The proposed model is still able to reduce the time complexity. I have some concerns related to motivation and the clarity of presenting or reasoning around some points that I explained in the weaknesses section.

---

> ### Author Response · Authors · 2022-11-09
> **Point-to-point response to comments from Reviewer DN9Y, part 1**
>
> > The motivation in some aspects is not clear. For example, the authors suggest the use of Recycling Positional Encoding to accommodate progressive caching of partial attention results for continual data streams. In section 3.7 they said that “There have been multiple prior works (Shaw et al., 2018; Huang et al., 2019; Dai et al., 2019) which create relative encodings by augmenting the SDA with positional offsets between query and keys. While such a modification to the continual attentions is possible, it hinders compatibility with the regular SDA.`` Why is Recycling Positional Encoding more helpful here and how does the previous model hinder the compatibility?
>
> A main motivation of the paper is to create a version of the original Transformer (Vaswani et al., 2017), that will be efficient during continual/online processing of tokens in a stream. As such, we are looking to derive an alternative module, which produces identical results to Eq. 1 of the paper.
>
> We have updated section 3.7 to clarify why it hinders compatibility: “There have been multiple prior works (Shaw et al., 2018; Huang et al., 2019; Dai et al., 2019) which create relative encodings by augmenting the SDA with positional offsets P between query and keys, i.e. A = exp(QK^T/√d + P ). While a similar modification to the continual attentions is possible, it is incompatibile with the regular SDA in Eq. (1). "
>
> In (Vaswani et al., 2017) the positional encodings are added prior to any Transformer Encoder layers. As such we can consider them external to the main architecture; changing the encoding doesn’t change its function. The issue with regular positional encoding for continual inference is described in the first paragraph of sec 3.7: “However, this static positional assignment is problematic in the context of continual inference; the last token at time t = 0 will be the next-to-last token at time t = 1, and thus in need of a different positional encoding than in the prior time-step.”.  A Recycling Positional Encoding scheme alleviates this issue.
>
> > The authors said that the proposed model works better with shallow architecture; is it still applicable to deeper architecture?
>
> We addressed this in Section 3.8 §Block count: “Nevertheless, it can be used in conjunction with a Continual Single-Output Transformer Encoder with optional regular Transformer Encoder blocks in between as illustrated in Fig. 1. The Regular Transformer Encoder blocks in the resulting architecture have a significantly larger computational complexity than the Continual Retroactive and Single-Output blocks”.
>
> So yes, it can work with deeper architectures. As we described in Section 3.8 §Block count, when we increase the number of transformer encoder blocks to  three and above, the efficiency gains get much smaller. This is because “middle layers” cannot be made continual. Only the first and last block lend themselves to ``continual'' of acceleration (see Figure 1). For computer vision problems it has been shown that using one-two blocks is sufficient to achieve state-of-the-art performance (see refs [8-11]).
>
> > How is the information accumulated from other tokens prior to classification?
>
> As our proposed Continual Transformer Encoder performs an identical computation to that of the original Transformer Encoder (Vaswani et al., 2017), the accumulation of knowledge from other tokens happens in the same way: through interactions in the (Recycling/Single-output) Scaled Dot-Product Attention. Eq. 8 states the correspondence between our proposed SDAs and the SDA in (Vaswani et al., 2017).
>
> Moreover, we conducted an ablation of the CLS token and it’s positions (see Table 1 (a)), where we found it to not be necessary in achieving good performance.

---

> > ### Comment · Reviewer_DN9Y · 2022-11-28
> > **Acknowledgment of Authors' feedback**
> >
> > The feedback from the Authors clears up most of the concerns, and I feel positive about the contribution. I have updated my rating accordingly.

---

> ### Author Response · Authors · 2022-11-09
> **Point-to-point response to comments from Reviewer DN9Y, part 2**
>
>
> > Uni-directional Recurrent Neural Networks are an example of Continual Inference Networks. How is the proposed model improved upon the Uni-directional Recurrent Network? For example, in SDA prior step features must be cached, re-loaded, and re-processed by the transformer in each step in correspondence with a predefined window-size of prior steps. Don't the authors think this is very complicated compared with RNN?
>
> We would argue that RNNs aren’t less complicated in general; for instance, the LSTM has forget, update, and output gates as well a hidden state, which the user is not exposed to during regular use. Similarly, the details of the (Continual) Transformer Encoders are packaged away, leaving a simple interface: Input a (sequence) of tokens and receive a (sequence) of tokens.
>
> Moreover, RNNs don’t seem to be able to provide the same performance level as Transformers. This is because the continual (and standard) Transformer Encoder has self-attention mechanism, which has been shown to improve performance considerably compared to RNNs with or without attention across many tasks. For instance, TRN in Table 2 (which is an RNN) achieves much lower mAP/mcAP compared to Transformer-based methods, while still requiring a large number of FLOPs per step prediction.
>
> > The experiments show that the proposed model performs well in time and memory reduction but it does not outperform SOTA. Can the authors explain more about that?
>
> While it is true that the predictive performance alone doesn’t surpass prior SotA methods, Continual Transformers provide far-superior computational efficiency. This is because the Continual Transformer Encoder perform the same calculations as regular Transformer Encoders (Viswani et al., 2017) but, in the Continual version, redundant computations over temporal sequences are avoided.
>
> On the pareto-frontier of predictive performance to FLOPs (see Fig. 2, where results closest to the upper left corner are pareto-optimal) the method provides SotA results by a large margin.
>
> For instance, the CoOadTR-b1 uses 237x less FLOPs than OadTR while retaining THUMOS14 mAP on Kinetics features and 255x less with 0.9 lower mAP for ActivityNet features. We have highlighted these metrics in Sec. 4.1.3.
>
> Here, we need to state again, that the contribution of the paper is the formulation of the continual inference network version of the Transformer Encoder (Viswani et al., 2017). Thus, the performance gains should not be considered only w.r.t. predictive accuracy, but mostly w.r.t. computational efficiency.
>
> > The paper is well written and is clear to follow. However, I found that there is a lack of motivation behind some points, such as the motivation for using Recycling Positional Encoding, and architecture modification. This makes it hard to draw conclusions from experiments.
>
> We have updated the paper (Sections 3.7 and 4.1.3) in order to cover the points mentioned by the reviewer’s comments. We hope that this makes the motivation behind the noted points clear.

---

### Official Review · Reviewer_bhNe · 2022-10-30

**Confidence:** 2
**Correctness:** 4
**Technical Novelty And Significance:** 4
**Empirical Novelty And Significance:** 4
**Recommendation:** 8

**Clarity, Quality, Novelty And Reproducibility:**

The paper is written in a clear manner.
The idea is novel.
Formulas and explanations in section 3 should be enough to reproduce the implementation and the results.

**Strength And Weaknesses:**

Strenghts:
 - novelty: to my knowledge there are no previous works trying to optimize computation inside the transformer architecture for online predictions
 - well written paper
 - fair discussion of the empriical results and of the limitations (the new architecture benefits more shallow models)

**Summary Of The Paper:**

The paper proposes an architectural modification of the transformer module, suitable for online predictions. The modification is meant to prevent heavy re-computations when new tokens arrive.

Authors propose two flavors of the continual transformer: one in which previous outputs are updated (revised), and one focused only on the prediction corresponding to the new token.

The paper clearly explains the computational gains of the continual transformer and tests it on a battery of online prediction datasets. The results show the benefits of the continual transformers on tasks that require shallow architectures.

**Summary Of The Review:**

Online prediction is crucial for a large set of problems, therefore the work is novel and of interest, and I suggest for the paper to be accepted.

---

### Decision · Program_Chairs · 2023-01-20

**Decision:**

Accept: poster

**Justification For Why Not Higher Score:**

The paper applies a technique that is quite similar to existing efficiency techniques (in particular caching during decoding and incremental attention computations) to a new domain. The score is not higher because that connection was not emphasized sufficiently in the current version.

**Justification For Why Not Lower Score:**

The authors introduce a meaningful efficiency gain for online data and the paper was universally supported by the reviewers.

**Metareview: Summary, Strengths And Weaknesses:**

This paper introduces an efficiency enhancement for transformers in the setting of online data. In particular, by caching prior activations and attention matrices, , reduces the computational complexity compared to recomputing the whole attention computation. The reviewers were in agreement that this contribution was worthy of ICLR publication. In preparing the camera ready version of this paper I would encourage the authors to emphasize relation to prior work where activations are cached during decoding (e.g. huggingface implementation)[https://github.com/huggingface/transformers/blob/d822ab636b6a14ed50f7bca0797c1de42c19de61/src/transformers/modeling_bart.py#L120-L122) and memory efficient attention implementations (eg  https://arxiv.org/abs/2112.05682). The first, a standard decoding implementation, and the latter, memory efficient attention implementations were not written with the online data setting in mind, but are examples of efficiency gains in the context of incrementally processing tokens and are worth discussing.

**Note From Pc:**

if the above contains the word "oral" or "spotlight" please see: "oral" presentation means -> notable-top-5% and "spotlight" means -> notable-top-25%. As stated in our emails, we are disassociating presentation type from AC recommendations